# Investigating Bioaccessibility of Advanced Glycation Product Precursors in Gluten-Free Foods Using In Vitro Gastrointestinal System

**DOI:** 10.3390/medicina59091578

**Published:** 2023-08-30

**Authors:** Yeliz Serin, Gamze Akbulut, Mustafa Yaman

**Affiliations:** 1Department of Nutrition and Dietetics, Faculty of Health Sciences, Cukurova University, 01380 Adana, Turkey; 2Department of Nutrition and Dietetics, Faculty of Health Sciences, Gazi University, 06490 Ankara, Turkey; gakbulut@gazi.edu.tr; 3Department of Nutrition and Dietetics, Faculty of Health Sciences, Istanbul Kent University, 34433 Istanbul, Turkey; 4Department of Nutrition and Dietetics, Faculty of Health Sciences, Istanbul Sabahattin Zaim University, 34303 Istanbul, Turkey

**Keywords:** advanced glycation end products, AGEs, bioaccessibility, gluten-free diet, glycation, glyoxal, in vitro, methylglyoxal

## Abstract

*Background and Objectives:* Gluten-free products have been produced as part of medical therapy and have gained popularity among individuals seeking weight loss or healthier dietary options. Assessing the potential risks associated with these products is essential in optimizing their compositions and developing new dietetic approaches. This study aimed to determine the glyoxal (GO) and methylglyoxal (MGO) contents in gluten-free bread, biscuits, and cookies and to examine their bioaccessibility using an in vitro gastrointestinal model. *Materials and Methods:* A total of 26 gluten-free and 19 gluten-containing (control) products were analyzed for their GO and MGO levels utilizing a high-performance liquid chromatography (HPLC) device. *Results:* Post-digestion, the GO and MGO values increased significantly across all food groups compared with pre-digestion values (*p* < 0.05), and the bioaccessibility exceeded 100%. Specifically, gluten-free bread exhibited higher post-digestion GO and MGO values than the control group (*p* < 0.05). Conversely, gluten-containing biscuits and cookies had higher post-digestion GO and MGO values compared to gluten-free products (*p* < 0.05). *Conclusions:* The detection of precursors to advanced glycation end products (AGEs) in gluten-free products has drawn attention to the potential health risks associated with their consumption. Therefore, reevaluation of the formulations and technologies used in these products and the introduction of new strategies are crucial in mitigating AGE content.

## 1. Introduction

Glycation refers to the nonenzymatic and spontaneous reaction of amino groups in proteins, nucleic acids, and lipids with glucose and other reducing sugars. In this process, the initial products in the ketoamine structure formed by glycation are unstable and undergo decomposition through oxidative and nonoxidative mechanisms called Maillard reactions (MR) [1]. The MR chains initiate with the condensation of a carbonyl group with an amino group, forming a Schiff base. The thermodynamically unstable Schiff base undergoes rearrangement to produce more stable ketoamines known as Amadori products. In the final stage of MR, Amadori products degrade to α-dicarbonyl compounds (α-DCs) such as glyoxal (GO) and methylglyoxal (MGO). Well-known advanced glycation end products (AGEs) include N-carboxymethyl lysine, N-carboxyethyl lysine, methylglyoxal–lysine dimers, pentosidine, and pyrraline [2,3]. AGEs are classified as endogenous and exogenous sources. The primary source of exogenous AGEs is one’s diet [4]. The rate and variety of AGE formation in foods are influenced by factors such as nutrient composition, the presence of advanced glycation precursors, the presence of transition metals, the water content, moisture, and pH of the environment, the heat treatments/duration of the heat treatment in the foods, and the presence of pro- and antioxidants [5]. A Western-style diet (processed foods, high-fat or sugary foods/drinks) is a primary source of dietary AGE intake [6]. On the contrary, endogenous AGEs are of biological origin and are formed during physiological glycation processes in organs, tissues, and body fluids [7].

Excessive production of advanced glycation products in the metabolism can trigger the activation of oxidative stress and inflammatory pathways, both of which are known to play significant roles in the pathogenesis of numerous diseases [8]. When AGEs bind to cell surfaces or cross-link with intra/extracellular proteins, they can induce alterations in the protein matrix and/or function. For instance, when AGEs modify intracellular mitochondrial proteins, they can lead to abnormal electrolyte transport and increased production of reactive oxygen species in the cell, ultimately resulting in mitochondrial dysfunction [9]. Moreover, AGEs can initiate pathogenic mechanisms by binding to various receptors in some organs and tissues. Among these receptors, the activation of Receptor-AGE (RAGE) stands out, as it can activate some inflammatory pathways by stimulation of NF-kB formation and reactive oxygen species release [8].

Human nutrition experiments can be ethically controversial, expensive, and often have high equipment requirements. On the other hand, in vitro simulated gastrointestinal digestive systems are widely utilized in laboratory settings across various fields of nutritional science research. In nutritional sciences, in vitro studies are commonly designed to investigate the bioaccessibility of macro- and micronutrients, leading to the development of new theories and hypotheses [10]. The assessment of food bioaccessibility is carried out using in vitro gastrointestinal systems. Bioaccessibility refers to the portion of an ingested biological component that becomes accessible for absorption through the epithelial layer of the gastrointestinal tract (GIT). For the biocomponents in food to be absorbed, they must first be separated from the food matrix and dissolved within the micellar structure. Bioavailability is expressed as a percentage and calculated as the ratio between the amount of biocomponent dissolved in the micellar phase and the amount of raw digested biocomponent [11].

Gluten, the primary storage protein of wheat, barley, and rye, generally affects the elasticity, viscosity, water absorption capacity, and structural properties of food products [12]. In addition, it contains proline-rich polypeptides resistant to digestive enzymes. These structural features of gluten can trigger autoimmune reactions in genetically susceptible individuals. This can lead to various gluten-related diseases such as celiac disease, dermatitis herpetiformis, wheat allergy, non-celiac gluten sensitivity, and contact urticaria, with celiac disease being the most common [13]. For individuals with gluten-related diseases, especially celiac disease, the mainstay of treatment is adhering to a gluten-free diet. The principle of this diet is based on eliminating wheat, barley, rye, and all gluten-containing products obtained from these raw materials [14]. In addition, the number of individuals consuming gluten-free products has increased due to concerns about healthy eating or weight loss [15]. While the literature often highlights the health benefits of gluten-free products in celiac treatment, potential risks are also emphasized, such as nutrient deficiencies or the unbalanced distribution of macro- or micronutrient compositions [14]. With an improved quality of life and the demand for quality foods, consumers not only are concerned with the color, aroma, and taste of foods but also attach great importance to the quality and safety of food [15]. Consequently, new approaches have been used to improve food quality and reduce potentially hazardous substances that may arise during food processing, especially in the production stages of the food industry [16]. Gluten-free products have been produced as a part of medical therapy for some time, but they have become popular recently among individuals seeking weight loss or healthier food choices [15]. Gluten-free bread, biscuits, and cookies are among the leading industrial components of the gluten-free diet [17]. Hence, it is vital to identify the potential risks and health benefits of these products and consider modifications to compositions or the formulation of new dietary approaches. In recent years, attention has been directed toward the potential health risks associated with the presence of AGEs in foods, leading to the development of new approaches to reduce AGEs [16].

This study aimed to determine the glyoxal (GO) and methylglyoxal (MGO) contents in gluten-free bread, biscuits, and cookies and to examine their bioaccessibility using an in vitro gastrointestinal model.

## 2. Materials and Methods

### 2.1. Sampling

This study involved different gluten-free products and their equivalent gluten-containing control group products purchased from markets in Istanbul and Ankara, Turkey, between February and March 2021. The sample size was determined based on the principles of determining nutritional components outlined by the European Food Information Resource [18], aiming to include at least 10 samples in each group. However, due to the varying market availability and stock status of gluten-free products, the study included 10 types of gluten-free bread, 5 types of gluten-free biscuits, and 11 varieties of gluten-free cookies. The control groups consisted of 9 types of bread, 5 types of biscuits, and 5 varieties of cookies, all containing gluten. Each sample in gluten-free groups was assigned a unique number code, which corresponded to the same samples in the control groups. The detailed ingredients of the selected products can be found in the Appendix A.

### 2.2. Chemicals

Various chemicals including GO (40%), MGO (40%), methanol, hydrochloric acid, acetonitrile, sodium hydroxide, 4-nitro-1,2-phenylenediamine, sodium acetate (CH_3_COONa), pancreatin (8 × USP specifications from pig pancreas), lipase (Type II from pig pancreas, 100–500 U/mg protein), alpha-amylase (from *Aspergillus oryzae* powder, 1.5 U/mg), pepsin (from pig gastric mucosa solid, lyophilized powder, 250 U/mg), mucin, NaCl, KCl, CaCl_2_·2H_2_O, NaHCO_3_, urea, bovine serum albumin, uric acid, bile salts, and other chemicals were obtained from Sigma–Aldrich (St. Louis, MO, USA).

### 2.3. Sugar Analysis

First, 45 products (26 gluten-free and 19 gluten-containing) were homogenized in a grinder (Sinbo SCM-2934, Istanbul, Turkey) for an average of 20–30 s. Then, 5 g of finely ground samples was weighed and placed in 50 mL falcon tubes. The volume was adjusted to 50 mL by adding distilled water, and the samples were homogenized in an ULTRA-TURRAX mixer (IKA, Staufen, Germany) for 1 min. After centrifugation at 13,000 rpm for 5 min, the samples were filtered through a 0.45 µm cellulose acetate membrane and analyzed using a HPLC system. The system consisted of a Shimadzu RI-20A detector (Shimadzu Corporation, Kyoto, Japan) and a Shimadzu LC 20AT pump. The flow rate was set to 2 mL/min using a mobile phase comprising acetonitrile and deionized water (80:20 *v*/*v*). Separation was carried out using an Agilent NH2 column (250 × 4.6 mm^2^, 5 μm) with the column furnace temperature set to 40 °C [19].

### 2.4. In Vitro Study

The bioaccessibility of GO and MGO in samples was determined using an in vitro human digestive system based on the method described by Yaman et al. (2022) [20]. The enzymes; organic and inorganic compounds; in vitro mouth, stomach, and small intestine environment; and bile solutions were prepared (Figure 1).

The digestive system consisted of three stages: the mouth, the stomach, and the small intestine. In the first step, 5 g of each sample was mixed with 5 mL of oral medium solution and incubated at 37 °C for 5 min in a shaking water bath. In the second step, the stomach medium was added to the orally digested sample and incubated at 37 °C for 2 h. In the third step, 10 mL of small intestine medium and 5 mL of bile solution were added to the stomach-digested mixture. The pH was adjusted to 7, and the sample was incubated for another 2 h at 37 °C in the shaking water bath. Following the final digestion phase, trichloroacetic acid was added to stop the digestion, and the test sample volume was adjusted to 50 mL of deionized water. The samples were then centrifuged at 8000 rpm for 10 min, and 10 mL of each supernatant was mixed with 10 mL of metaphosphoric acid solution. The solution was subsequently filtered using a cellulose acetate (CA) filter (0.45 µm).

#### 2.4.1. Extraction of GO and MGO

The extraction of GO and MGO components from the foods followed the procedure described by Cengiz et al. [20]. Initially, 5 g of each sample was homogenized in a grinder (Sinbo SCM-2934, Turkey) for 20–25 s. Then, 5 g of the ground samples was weighed and transferred into 50 mL falcon tubes. To these, 25 mL of methanol was added, and the mixtures were subjected to extraction using an ULTRA-TURRAX device (IKA, Staufen, Germany) at 15,000 rpm for 5 min. Subsequently, 0.5 mL of the supernatant was pipetted and transferred to 10 mL glass tubes, and 2 mL of the prepared sodium acetate solution was added. Further, 0.5 mL of 4-nitro-1,2-phenylenediamine derivatization solution was added, and the mixture was incubated in a water bath at 70 °C for 20 min. After derivatization, the mixture was filtered through a 0.45 µm CA filter and injected into the HPLC device. All samples were prepared under standard conditions, considering environmental factors (e.g., high temperature, UV light) that might cause oxidation during sample preparation.

#### 2.4.2. HPLC Analysis of Samples

The HPLC system consisted of a Shimadzu LC 20AT pump with a Shimadzu SPD-20A UV/VIS detector (Shimadzu Corporation, Kyoto, Japan). The mobile phase comprised methanol:water:acetonitrile in a ratio of 42:56:2 (*v*/*v*/*v*), and the experiment was performed at a wavelength of 254 nm. The GO and MGO were separated using an Inertsil ODS-3 column (250 × 4.6 mm^2^, 5 μm) with a flow rate of 1 mL/min. The column oven temperature was maintained at 25 °C. The HPLC chromatogram of GO and MGO, which were advanced glycation precursors analyzed in this study (gluten-free sample 4), is shown in Figure 2.

The method validation of GO and MGO analysis was based on modifications made to an existing procedure according to Association of Official Agricultural Chemists (AOAC) guidelines [21]. This analysis method was validated previously by Cengiz and Yaman et al. [20,22].

### 2.5. Statistical Analysis

The data obtained from the triple HPLC analysis of each gluten-free and control group product were analyzed using SPSS (22.0) and Minitab programs. Intra-group comparisons (before and after digestion) were made using analysis of variance and post hoc Tukey tests in the Minitab statistical program. Comparisons between groups were analyzed with the SPSS (22.0) package program. For comparisons between groups, the conformity of the variables to the normal distribution was examined visually (histogram and probability graphs) and analytically (Shapiro–Wilk test). The independent-samples *t* test was used for normally distributed data, whereas the Mann–Whitney *U* test was used for non-normally distributed data. Descriptive variables were expressed as mean ± standard deviation. A *p* value < 0.05 was considered statistically significant in both statistical programs.

## 3. Results

The results of the sugar analysis (glucose, fructose, and sucrose) for the products are presented in Table 1. The comparison of sugar composition between gluten-free and gluten-containing products did not show any significant difference (*p* > 0.05, Table 1).

The bioaccessibility values of GO and MGO in the gluten-free and gluten-containing bread before and after digestion are summarized in Table 2. In all products, the GO and MGO values and bioaccessibility after digestion increased compared with pre-digestion values (*p* < 0.05).

The bioaccessibility values of GO and MGO in the gluten-free and gluten-containing biscuits before and after digestion are summarized in Table 3. The post-digestion GO and MGO values in all gluten-free and control group biscuit varieties increased compared with the pre-digestion values. The bioavailability of GO and MGO after digestion was more than 100% in all product groups (Table 3; *p* < 0.05).

The bioaccessibility values of GO and MGO in the gluten-free and gluten-containing cookies before and after digestion are summarized in Table 4. The post-digestion GO and MGO values in gluten-free and gluten-containing cookies were higher than their pre-digestion values (Table 4; *p* < 0.05). The GO and MGO bioaccessibility values were above 100% in gluten-free and control groups.

## 4. Discussion

The majority of gluten-free and gluten-containing packaged products fall into the category of processed/ultra-processed foods, as per the NOVA food classification system [23]. Processed/ultra-processed foods are industrial products that undergo minimal or no pre-consumption processing [24]. The composition and macronutrient profiles of these industrial products can vary depending on the formulations used by the manufacturer and the substances added during the production phase. Based on the label information of the products assessed in this study, the main carbohydrate sources in gluten-free products are rice flour and corn starch, while the oil sources are palm and sunflower. In contrast, gluten-containing products primarily use wheat flour as their carbohydrate source (Appendix A).

AGEs are known to form easily through sugar autoxidation in industrial foods. Despite limited data on how sugar type influences the formation of AGEs, fructose has been identified as having the maximum influence on the formation of GO and MGO [7,20]. In this study, the amount of sucrose in the products analyzed was higher than that of glucose and fructose. However, no significant difference was observed in sugar composition between the gluten-free and gluten-containing food groups (*p* > 0.05; Table 1).

The formation of GO and MGO in bread prior to digestion is mainly associated with the low moisture level of bread and the high heat treatment it undergoes during baking. During baking, the surface temperature of the dough reaches about 230–250 °C, leading to the evaporation of water and the browning of the bread. The reduction in moisture levels and exposure to high temperatures, especially regarding bread’s outer crust, create ideal conditions for Maillard reactions to occur. Maillard reactions (cooking time, temperature, and storage conditions), in turn, contribute to the formation of AGEs [2,5]. In this study, the pre-digestion MGO levels of gluten-free samples were higher than those of gluten-containing products (Table 2; *p* < 0.05). The MGO levels tend to be higher in thermally processed foods, especially in bakery and fried products [25]. Most gluten-free baked products often require multiple processing steps to achieve the same texture and acceptability as their equivalent products [26]. These steps may be directly related to the MGO levels of gluten-free products.

The post-digestion GO–MGO levels and bioaccessibility of foods are influenced by the composition and starch structure of the food. The heat treatment applied during food processing also plays a role in determining digestibility [5,27]. Evaluating the increase in the number and bioaccessibility of AGEs in the digestive environment is primarily essential for evaluating health-related metabolic processes [28]. In this study, the post-digestion GO levels of gluten-free products were found to be higher than those of gluten-containing foods (Table 2; *p* < 0.05). This difference might be attributed to the distinct ingredients used in samples (Appendix A). Primary carbohydrate sources of gluten-free products predominantly rely on corn or rice starch [26]. These starch types are composed of amylose and amylopectin, with amylose being more resistant to digestive enzymes than amylopectin. Corn and rice varieties typically contain very little amylose, while wheat starch contains 23–27% amylose [29]. The proportional difference of amylose/amylopectin may influence digestive enzyme activity and indirectly influence the number of AGEs that are released from food.

The higher level of polyunsaturated fatty acids in foods is a factor in increasing lipid oxidation and AGE formation [30]. It has been reported that gluten-free products have higher fat content than gluten-containing equivalents [31,32,33]. The fat content of gluten-free products analyzed in this study was also higher than that of those containing gluten. (Appendix A).

Staling may be another factor that affects the number of AGEs in food. The fast staling of gluten-free bread is primarily attributed to the starch structure, which is the raw material used. A study has shown that rice-based bread types become stale faster than wheat-type bread [34]. The lower contents of amylose and amylopectin in gluten-free bread compared with wheat flour are another reason for staleness [35]. Therefore, excessive starch degradation (staling) is a factor that accelerates Maillard reactions and affects the formation of AGEs [36]. The GO bioaccessibility of white bread is less than 77% compared with other bread types. This is because gluten-containing white bread is a commonly produced and consumed food in bakeries in our country, resulting in a shorter shelf life compared with packaged bread. On the contrary, gluten-free bread is usually preferred by individuals following a gluten-free diet; thus, we expect a longer shelf life for gluten-free bread. However, this extended shelf life may contribute to an increase in AGE content in gluten-free breads [37].

Biscuits are popular in the food industry due to their variable taste, aroma, texture, long shelf life, and high consumer demand. The essential nutritional components of biscuits include flour, vegetable or saturated fat, sugar, water, and some chemical components [38]. Moreover, biscuits are considered one of the primary dietary sources of α-dicarbonyl compounds owing to varying amounts of high-fructose corn syrup present in them [39]. Previous studies reported that the average MGO value in widely consumed biscuit types was 39.64 ± 4.57 μg/100 g [39]. Another study found the GO and MGO values in biscuits to be 35–224 µg/100 g and 32–1573 µg/100 g, respectively [40]. In the present study, the pre-digestion GO values in the two groups were found to be similar (*p* > 0.05; Table 3). The formation of GO in biscuits primarily stems from sugar autoxidation [41]. The lack of difference in GO formation may be attributed to the similarity in sugar composition between the gluten-free and gluten-containing samples (Table 1, *p* > 0.05).

The increased GO and MGO values in biscuits are influenced by various factors, such as reduced sugar and unsaturated fatty acids in the structure of biscuits, lipid oxidation, caramelization, and Maillard reactions [42]. In addition, high temperature, low water activity, baking time, moisture, and pH are critical factors affecting AGE formation in biscuits [40]. The interactions between food ingredients in the digestive environment can contribute to increased AGE formation and their bioaccessibility (Appendix A Appendix A). In this study, post-digestion GO and MGO values in gluten-free products were found to be higher than those in gluten-containing products (*p* < 0.05; Table 3). This difference might be attributed to the lower protein levels of gluten-free products compared to those of gluten-containing products [26]. Amino acids can efficiently scavenge methylglyoxal, leading to the formation of numerous adducts due to their highly nucleophilic reactivity with methylglyoxal [25].

Cookies are composed of various ingredients such as sugar, protein, fat, eggs, and nuts (Appendix A). During production, they are subjected to high-temperature and dry-heat cooking processes, making them an ideal model for investigating chemical hazards derived from the Maillard reaction, such as acrylamide and advanced glycation products [5]. The GO and MGO values of cookies range from 4.8 to 26.0 mg/kg and from 3.7 to 81.4 mg/kg, respectively [42]. In this study, the pre-digestion median GO and MGO values of gluten-free cookies were determined as 25.0 (22.0) µg/100 g and 68.0 (51.0) µg/100 g, respectively (Table 4). The differences in results from various analysis methods are often due to the variability in laboratory conditions and the calculation techniques used when quantifying AGEs. Comparisons between different analytical methods can be challenging due to the need for unit conversion, which may lead to inconclusive results [43,44].

The increase in GO and MGO values in cookies after digestion is mainly associated with Maillard reactions, oxidation of sugars, caramelization, and lipid oxidation [5]. The high sugar and fat contents of these products, as well as the high content of components such as butter, egg, and milk powder, may contribute to the synthesis of endogenous GO and MGO [38]. The exposure of foods containing protein and carbohydrates to high heat is among the leading cause of Maillard reactions responsible for food browning [29]. After digestion, the GO and MGO values tended to decrease compared with the initial values in the cookies with sample code 22 from the control group (Table 4; *p* < 0.05). This decrease might be attributed to the product composition of the cookie containing grapes. Studies have reported that red grape seed extract reduces precursors of advanced glycation products. The total phenolic compounds, flavonoids, and anthocyanins in grapes may decrease glycation and oxidation, leading to GO and MGO formation [45,46,47]. This finding highlights the importance of selecting raw materials with potential antiglycation properties while developing products to reduce AGE formation.

The pre-and post-digestion GO and MGO values of the gluten-containing cookies were higher than those of the gluten-free cookies (Table 4; *p* < 0.05). This difference can be attributed to food compositions, which influence the amount of AGE formation. GO is formed endogenously mainly by the autoxidation of sugars or the peroxidation of polyunsaturated fatty acids [48]. In particular, the polyunsaturated fatty acid ratio of sunflower oil is approximately 66%, and higher numbers of double bonds in this fatty acid correspond to higher rates of oxidation [29]. Therefore, the increase in the oxidation of sugars and fats in gluten-containing products in vitro may affect the formation of AGEs.

## 5. Conclusions

This study was novel in determining the AGE precursors of gluten-free products, comparing them with gluten-containing products, and evaluating their effects in the in vitro digestive environment. The findings from this investigation revealed that both gluten-free and control group products had increased GO, MGO, and bioaccessibility values in the digestive environment. In addition, gluten-free bread exhibited higher GO and MGO values compared with gluten-containing bread.

The significance of this research lies in its determination of the potential health risks associated with gluten-free products, which are an essential part of the gluten-free diet [49]. When producing these products, production technology and product formulations should be reconsidered. In future studies, it is recommended to assess the influence of factors such as nutrient components (e.g., amino acid, glucose, fructose, fatty acids) and their interactions in the digestive environment on AGE formation. Additional efforts should be made to develop new product formulations aimed at reducing the amount of AGE precursors.

## Figures and Tables

**Figure 1 medicina-59-01578-f001:**
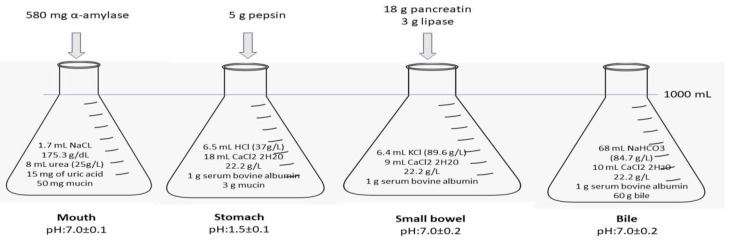
In vitro digestion environment.

**Figure 2 medicina-59-01578-f002:**
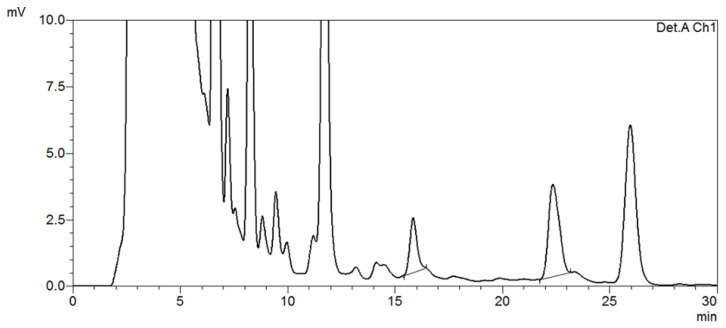
HPLC chromatogram of GO and MGO.

**Table 1 medicina-59-01578-t001:** Comparison of the simple sugar content of gluten-free products and control groups.

	Fructose		Glucose		Sucrose	
Variables	Mean ± SD	Median (IQR) Min–Max	Mean ± SD	Median (IQR) Min–Max	Mean ± SD	Median (IQR) Min–Max
Gluten-free bread (*n* = 10)	0.4 ± 0.2	0.3 (0.5) 0.0–0.8	0.4 ± 0.4	0.3 (0.7) 0.0–1.0	1.9 ± 5	0.0 (0.7) 0.0–17.0
Control bread (*n* = 9)	0.2 ± 0.3	0.0–0.87 0.0 (0.6)	0.1 ± 0.3	0.0–0.7 0.0 (0.5)	0.0 ± 0	0.0–0.0 0.0 (0.0)
*p* value	0.11		0.23		0.54	
Gluten-free biscuit (*n* = 5)	0.3 ± 0.2	0.3 (0.6) 0.0–0.7	3.9 ± 5.1	0.6 (9.0) 0.0–11.2	12.9 ± 12	18.5 (23.2) 0.0–23.9
Control biscuit (*n* = 5)	1.2 ± 1.4	0.7 (1.6) 0.0–3.5	1.0 ± 0.9	0.8 (1.7) 0.0–2.3	22.1 ± 15	23.6 (23.5) 0.0–39.9
*p* value	0.30		0.84		0.31	
Gluten-free cookie (*n* = 11)	0.4 ± 0.9	0.0 (0.5) 0.0–3.2	0.5 ± 1.3	0.0 (0.0) 0.0–4.3	19.5 ± 8	20.0 (10.3) 2.3–29.7
Control cookie (*n* = 9)	0.5 ± 0.5	0.3 (1.1) 0.0–1.3	0.5 ± 0.6	0.2 (1.5) 0.0–1.7	25.3 ± 6	0.3 (1.9) 0.0–3.9
*p* value	0.12		0.10		0.14	

Parametric data: Independent-samples *t* test; nonparametric data: Mann–Whitney *U* test. IQR, Interquartile range.

**Table 2 medicina-59-01578-t002:** Comparison of mean glyoxal, methylglyoxal, and bioaccessibility of gluten-free and control group bread before and after digestion.

Gluten-Free	GO (µg/100 g) Mean ± SD		MGO (µg/100 g) Mean ± SD		Bioaccessibility (%)	
	Pre-Digestion	Post-Digestion	Pre-Digestion	Post-Digestion	GO	MGO
1	17 ± 2 ^a^	223 ± 7 ^b^	72 ± 4 ^a^	537 ± 25 ^b^	1288	746
2	22 ± 1 ^a^	154 ± 4 ^b^	81 ± 4 ^a^	230 ± 7 ^b^	691	282
3	17 ± 1 ^a^	157 ± 5 ^b^	68 ± 2 ^a^	149 ± 5 ^b^	944	218
4	46 ± 2 ^a^	179 ± 5 ^b^	233 ± 11 ^a^	407 ± 18 ^b^	390	175
5	51 ± 3 ^a^	246 ± 12 ^b^	227 ± 11 ^a^	449 ± 20 ^b^	485	198
6	18 ± 2 ^a^	130 ± 6 ^b^	43 ± 2 ^a^	71 ± 4 ^b^	738	166
7	42 ± 2 ^a^	44 ± 2 ^a^	56 ± 3 ^a^	427 ± 19 ^b^	105	766
8	49 ± 2 ^a^	611 ± 16 ^b^	34 ± 2 ^a^	105 ± 4 ^b^	1247	305
9	170 ± 8 ^a^	222 ± 6 ^b^	442 ± 10 ^a^	502 ± 23 ^b^	130	114
10	147 ± 3 ^b^	279 ± 13 ^b^	378 ± 10 ^a^	914 ± 42 ^b^	189	242
* Median (IQR) Min–Max	44.0 (32.8) 16.0–178.0	199.5 (97.8) 42.0–626.0	76.0 (182.5) 33.0–444.8	416.0 (358.8) 68.8–954.0		
**Control**						
1, 2, and 3	209 ± 8 ^a^	161 ± 3 ^b^	18 ± 1 ^a^	161 ± 5 ^b^	77	878
4	31 ± 2 ^a^	166 ± 8 ^b^	130 ± 4 ^a^	304 ± 14 ^b^	540	234
5	17 ± 1 ^a^	43 ± 4 ^b^	22 ± 1 ^a^	98 ± 6 ^b^	255	444
6	41 ± 3 ^a^	182 ± 8 ^b^	257 ± 5 ^a^	544 ± 25 ^b^	448	212
7	11 ± 1 ^a^	135 ± 6 ^b^	211 ± 8 ^a^	522 ± 29 ^b^	1269	248
8	49 ± 3 ^a^	169 ± 8 ^b^	59 ± 2 ^a^	274 ± 13 ^b^	342	467
9	23 ± 1 ^a^	101 ± 6 ^b^	183 ± 6 ^a^	244 ± 66 ^b^	434	134
* Median (IQR) Min–Max	35.5 (186.3) 0.0–219.0	158.0 (66.3) 0.0–190.0	40.0 (167.0) 0.0–263.0	167.5 (160.0) 0.0–568.0		
**Comparison of mean values of GO and MGO between groups**						
Gluten-free	57.9 ± 53	224.6 ± 14	163 ± 143	379 ± 245		
Control	80.4 ± 88	127.7 ± 59	91.5 ± 92	246 ± 169		
*p* value	0.95	0.001 **	0.01 **	0.11		

GO, glyoxal; MGO, methylglyoxal. Control: gluten-containing. Numerical values are shown as mean ± standard deviation of three replicate measurements. * Median and IQR (interquartile range) values. Different letters in the same rows indicate statistical differences in the pre- and post-digestion values of GO and MGO (ANOVA, post hoc Tukey tests, *p* < 0.05). ** *p* < 0.05, Mann–Whitney *U* test.

**Table 3 medicina-59-01578-t003:** Comparison of mean glyoxal, methylglyoxal, and bioaccessibility of gluten-free and control group biscuits before and after digestion.

	GO (µg/100 g) Mean ± SD		MGO (µg/100 g) Mean ± SD		Bioaccessibility (%)	
Gluten-Free	Pre-Digestion	Post-Digestion	Pre-Digestion	Post-Digestion	GO	MGO
11	17 ± 1 ^a^	401 ± 11 ^b^	68 ± 2 ^a^	831 ± 19 ^b^	2406	1228
12	19 ± 2 ^a^	313 ± 15 ^b^	55 ± 1 ^a^	165 ± 4 ^b^	1617	301
13	138 ± 7 ^a^	394 ± 18 ^b^	114 ± 4 ^a^	182 ± 8 ^b^	286	160
14	12 ± 1 ^a^	451 ± 18 ^b^	97 ± 3 ^a^	1778 ± 80 ^b^	3866	1839
15	80 ± 3 ^a^	169 ± 7 ^b^	68 ± 3 ^a^	92 ± 6 ^b^	212	135
* Median (IQR) Min–max	19.0 (67.0) 11.0–144.0	391.0 (116.0) 163.0–469.0	69.0 (35.0) 54.0–118.0	183.0 (689.0) 87.0–1855.0		
**Control**						
11	62 ± 2 ^a^	832 ± 33 ^b^	322 ± 7 ^a^	1562 ± 71 ^b^	1335	486
12	84 ± 4 ^a^	525 ± 12 ^b^	469 ± 22 ^a^	1145 ± 52 ^b^	627	244
13	94 ± 5 ^a^	281 ± 10 ^b^	374 ± 8 ^a^	399 ± 13 ^b^	298	107
14	26 ± 1 ^a^	830 ± 38 ^b^	200 ± 4 ^a^	2252 ± 102 ^b^	3152	1126
15	63 ± 4 ^a^	665 ± 30 ^b^	214 ± 12 ^z^	1327 ± 60 ^b^	1056	619
* Median (IQR) Min–max	64.0 (29.0) 25.0–99.0	667.0 (321.0) 271.0–866.0	320.0 (179.0) 196.0–490.0	1332.0 (538.0) 387.0–2350.0		
**Comparison of mean values of GO and MGO between groups**						
Gluten-free	53.1 ± 50	345.6 ± 102	80.2 ± 22	609.6 ± 665		
Control	65.9 ± 24.3	626.6 ± 215	315.8 ± 104	1337.2 ± 624		
*p* value	*0.12*	*0.001 ***	*< 0.001 ***	*0.004 ***		

GO, glyoxal; MGO, methylglyoxal. Control: gluten-containing Numerical values are shown as mean ± standard deviation values of three replicate measurements. * Median and IQR (interquartile range) values. Different letters in the same rows indicate statistical differences in the pre- and post-digestion values of GO and MGO (ANOVA, post hoc Tukey tests, *p* < 0.05). ** *p* < 0.05, Mann–Whitney *U* test.

**Table 4 medicina-59-01578-t004:** Comparison of mean glyoxal, methylglyoxal, and bioaccessibility of gluten-free and control group cookies before and after digestion.

	GO (µg/100 g) Mean ± SD		MGO (µg/100 g) Mean ± SD		Bioaccessibility (%)		
Gluten-Free	Pre-Digestion	Post-Digestion	Pre-Digestion	Post-Digestion	GO	MGO	
16	203 ± 9 ^a^	557 ± 17 ^b^	1346 ± 61 ^a^	2297 ± 104 ^b^	275	171	
17	28 ± 1 ^a^	137 ± 8 ^b^	85 ± 4 ^a^	188 ± 9 ^b^	482	220	
18	139 ± 7 ^a^	365 ± 13 ^b^	513 ± 24 ^a^	533 ± 24 ^a^	263	104	
19	18 ± 1 ^a^	76 ± 2 ^b^	79 ± 3 ^a^	381 ± 9 ^b^	420	480	
20	24 ± 2 ^a^	68 ± 2 ^b^	37 ± 2 ^a^	89 ± 2 ^b^	285	239	
21	12 ± 2 ^a^	46 ± 2 ^b^	68 ± 3 ^a^	120 ± 6 ^b^	370	176	
22	31 ± 2 ^a^	19 ± 1 ^b^	0 ± 0 ^a^	0 ± 0 ^a^	62	0	
23	14 ± 1 ^a^	51 ± 4 ^b^	34 ± 2 ^a^	125 ± 9 ^b^	356	371	
24	37 ± 1 ^a^	66 ± 2 ^b^	71 ± 2 ^a^	252 ± 8 ^b^	178	356	
25	15 ± 2 ^a^	70 ± 3 ^b^	34 ± 1 ^a^	55 ± 1 ^a^	477	162	
26	7 ± 1 ^a^	74 ± 4 ^b^	10 ± 2 ^a^	74 ± 5 ^b^	1110	762	
* Median (IQR) Min–max	25.0 (22.0) 6.0–211.0	70.0 (80.0) 18.0–568.0	68.0 (51.0) 0.0–1404.0	125.0 (299.0) 0.0–2397.0			
**Control**							
16 and 17	29 ± 1 ^a^	485 ± 19 ^b^	69 ± 4 ^a^	572 ± 26 ^b^	1672	825	
18	70 ± 2 ^a^	314 ± 15 ^b^	197 ± 9 ^a^	206 ± 16 ^b^	451	105	
19	43 ± 2 ^a^	139 ± 7 ^b^	99 ± 2 ^a^	293 ± 14 ^b^	322	297	
20	74 ± 3 ^a^	302 ± 14 ^b^	465 ± 21 ^a^	884 ± 35 ^b^	408	190	
21 and 22	140 ± 7 ^a^	207 ± 9 ^b^	556 ± 22 ^a^	560 ± 25 ^a^	148	101	
23	77 ± 3 ^a^	279 ± 13 ^b^	124 ± 4 ^a^	361 ± 16 ^b^	362	290	
24	413 ± 19 ^a^	110 ± 5 ^b^	335 ± 15 ^a^	101 ± 6 ^b^	27	30	
* Median (IQR) Min–max	74.0 (99.0) 27.0–431.0	280.0 (130.0) 106.0–504.0	198.0 (389.0) 65.0–578.0	534.0 (305.0) 96.0–919.0			
**Comparison of mean values of GO and MGO between groups**							
Gluten-free	49.7 ± 61	138.9 ± 162	207 ± 391	373.9 ± 636			
Control	112.7 ± 115.4	281.0 ± 129	274 ± 198	456.6 ± 230			
*p* value	*<0.001 ***	*<0.001 ***	*<0.001 ***	*<0.001 ***			

GO, glyoxal; MGO, methylglyoxal. Control: gluten-containing. Numerical values are shown as mean ± standard deviation values of three replicate measurements. ^*^ Median and IQR (interquartile range) values. Different letters in the same rows indicate statistical differences in the pre- and post-digestion values of GO and MGO (ANOVA, post hoc Tukey tests, *p* < 0.05). ** *p* < 0.05, Mann–Whitney *U* test.

## Data Availability

The authors confirmed that the data supporting the findings of this study are available within the article and its Appendix A. Raw data that support the findings of this study are available from the corresponding author, upon reasonable request.

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
