# Peer review of "Investigating Bioaccessibility of Advanced Glycation Product Precursors in Gluten-Free Foods Using In Vitro Gastrointestinal System"

_medicina, 2023, doi:10.3390/medicina59091578_

Round 1
Reviewer 1 Report
This manuscript determined the glyoxal (GO) and methylglyoxal (MGO) content of gluten-free breads, cookies and cookies and tests their bio acceptability in an in vitro gastrointestinal model. The detection of advanced glycation end-products (AGEs) precursors in gluten-free products has raised concerns about the potential health risks of these products. This article reevaluated the formulation and technology of these products and provides new strategies to reduce the levels of AGEs in order to change the product composition to develop new dietary approaches. However, this article still needs to undergo some revisions before being accepted.
1. Checking the key words, for example, advanced? I think advanced glycation end products would be more appropriate.
2. In line 54 and 95, in vitro should be revised to in vitro.
3. In lines 91-92, AGEs were already covered in the previous natural paragraph. This part would be more appropriately placed in the previous paragraph.
4. Is the internal or external standard method used for the application of HPLC to detect the content of AGEs? It should be elaborated in the method. Please refer this reference (Food chemistry, 385(2022): 132697. Food Bioscience, 43(2021), 101313).
5. Prospects and perspectives should be presented in the conclusion.
6. Reference formatting should be rechecked, e.g., at lines 388 and 390.
7. In lines 338, “This study is critical because it determines the potential health risks of gluten-free products which are an essential part of the gluten-free diet”. What's that supposed to mean? It is better to refer this reference (Journal of Food Processing and Preservation,2021, 45(9),e15684.).
8. The grammar and tenses of the entire manuscript should be rechecked.
This manuscript determined the glyoxal (GO) and methylglyoxal (MGO) content of gluten-free breads, cookies and cookies and tests their bio acceptability in an in vitro gastrointestinal model. The detection of advanced glycation end-products (AGEs) precursors in gluten-free products has raised concerns about the potential health risks of these products. This article reevaluated the formulation and technology of these products and provides new strategies to reduce the levels of AGEs in order to change the product composition to develop new dietary approaches. However, this article still needs to undergo some revisions before being accepted.
1. Checking the key words, for example, advanced? I think advanced glycation end products would be more appropriate.
2. In line 54 and 95, in vitro should be revised to in vitro.
3. In lines 91-92, AGEs were already covered in the previous natural paragraph. This part would be more appropriately placed in the previous paragraph.
4. Is the internal or external standard method used for the application of HPLC to detect the content of AGEs? It should be elaborated in the method. Please refer this reference (Food chemistry, 385(2022): 132697. Food Bioscience, 43(2021), 101313).
5. Prospects and perspectives should be presented in the conclusion.
6. Reference formatting should be rechecked, e.g., at lines 388 and 390.
7. In lines 338, “This study is critical because it determines the potential health risks of gluten-free products which are an essential part of the gluten-free diet”. What's that supposed to mean? It is better to refer this reference (Journal of Food Processing and Preservation,2021, 45(9),e15684.).
8. The grammar and tenses of the entire manuscript should be rechecked.
Author Response
We are thankful to the reviewers and editor for their valuable contributions. We have re-arranged and explained all the suggestions point by point.
REVIEWER 1
Comments and Suggestions for Authors
This manuscript determined the glyoxal (GO) and methylglyoxal (MGO) content of gluten-free breads, cookies and cookies and tests their bio acceptability in an in vitro gastrointestinal model. The detection of advanced glycation end-products (AGEs) precursors in gluten-free products has raised concerns about the potential health risks of these products. This article reevaluated the formulation and technology of these products and provides new strategies to reduce the levels of AGEs in order to change the product composition to develop new dietary approaches. However, this article still needs to undergo some revisions before being accepted.
- Checking the key words, for example, advanced? I think advanced glycation end products would be more appropriate.
The keywords have checked and re-arranged by alphabetical order. Advanced glycation end products and AGEs ,general uses of this term, added to the key words part.
- In line 54 and 95, in vitro should be revised to in vitro.
Thank you very much; It revised as your suggested.
- In lines 91-92, AGEs were already covered in the previous natural paragraph. This part would be more appropriately placed in the previous paragraph.
The expression is modified and fixed to the natural flow as ‘’ Therefore, it is essential to determine the potential risks and health benefits of these products, and modify their compositions or formulate new dietary approaches. In recent years, attention has been drawn to the potential health risks posed by the presence of AGEs in foods and attempts have been made by the food industry to develop new approaches to reduce AGEs ‘’
- Is the internal or external standard method used for the application of HPLC to detect the content of AGEs? It should be elaborated in the method. Please refer this reference (Food chemistry, 385(2022): 132697.Food Bioscience, 43(2021), 101313).
Thank you very much for this suggestion. The validation prosedures and steps have added at line 183-185.
- Prospects and perspectives should be presented in the conclusion.
Thank you very much.It has been re-organised again.
- Reference formatting should be rechecked, e.g., at lines 388 and 390.
All the references have checked again.
- In lines 338, “This study is critical because it determines the potential health risks of gluten-free products which are an essential part of the gluten-free diet”. What's that supposed to mean? It is better to refer this reference (Journal of Food Processing and Preservation,2021, 45(9),e15684.).
Thank you very much. This explanation is located in conclusion part, not in line 338. It has cited with suggested reference.
- The grammar and tenses of the entire manuscript should be rechecked.
The manuscript has checked by proof and editing service and the certificate of editing upload to the system
Reviewer 2 Report
The study focused on analyzing the levels of glyoxal and methylglyoxal in gluten-free baked goods such as bread, biscuits, and cookies and evaluating their bioaccessibility in simulated gastrointestinal conditions. It is interesting topic, however, there still have some issues need to check.
1. Please verify the keywords. It should be AGEs.
2. The Maillard reaction should be introduced in introduction. Please refer this reference (Food Chemistry, 417(2023):135861.).
3. AGEs metabolic pathways and potential risks need to be introduced and evaluated, which is the basis of this study (Critical Reviews in Food Science and Nutrition, Doi: 10.1080/10408398.2022.2076064.).
4. In Table 4. The Bio accessibility should be expressed with M±SD.
5. The grammar and tenses of the entire manuscript should be rechecked.
6. The reference should be update in recent years.
The study focused on analyzing the levels of glyoxal and methylglyoxal in gluten-free baked goods such as bread, biscuits, and cookies and evaluating their bioaccessibility in simulated gastrointestinal conditions. It is interesting topic, however, there still have some issues need to check.
1. Please verify the keywords. It should be AGEs.
2. The Maillard reaction should be introduced in introduction. Please refer this reference (Food Chemistry, 417(2023):135861.).
3. AGEs metabolic pathways and potential risks need to be introduced and evaluated, which is the basis of this study (Critical Reviews in Food Science and Nutrition, Doi: 10.1080/10408398.2022.2076064.).
4. In Table 4. The Bio accessibility should be expressed with M±SD.
5. The grammar and tenses of the entire manuscript should be rechecked.
6. The reference should be update in recent years.
Author Response
REVIEWER 2
Comments and Suggestions for Authors
The study focused on analyzing the levels of glyoxal and methylglyoxal in gluten-free baked goods such as bread, biscuits, and cookies and evaluating their bioaccessibility in simulated gastrointestinal conditions. It is interesting topic, however, there still have some issues need to check.
- Please verify the keywords. It should be AGEs.
The keywords have checked and re-arranged by alphabetical order and advanced glycation end products and AGEs ,general uses of this term, added to the key words part.
- The Maillard reaction should be introduced in introduction. Please refer this reference (Food Chemistry, 417(2023):135861.).
The reference has added after MR definition .
- AGEs metabolic pathways and potential risks need to be introduced and evaluated, which is the basis of this study (Critical Reviews in Food Science and Nutrition, Doi: 10.1080/10408398.2022.2076064.).
Thank you very much for the suggestion. AGEs metabolic pathways and potential risks have introduced and explained in line :45-52. Besides, the reference that you suggested also has added.
- In Table 4. The Bioaccessibility should be expressed with M±SD.
Bioaccessibility is the portion of an ingested biological component that becomes accessible for absorption through the epithelial layer of the gastrointestinal tract (GIT). For food to be absorbed, biocomponents must first be separated from the food matrix and dissolved in the micellar structure. Bioavailability is expressed as a percentage and calculated as the ratio between the amount of biocomponent dissolved in the micellar phase and the amount of raw digested biocomponent Line :58-64( Alminger, M., Aura, A. M., Bohn, T., Dufour, C., El, S. N., Gomes, A., Karakaya, S., Martínez-Cuesta, M. C., McDougall, G. J., Requena, T. and Santos, C. N. (2014). In vitro models for studying secondary plant metabolite digestion and bioaccessibility. Comprehensive Reviews in Food Science and Food Safety, 13(4), 413-436.)
- The grammar and tenses of the entire manuscript should be rechecked.
- The manuscript has checked by proof and editing service and the certificate of editing upload to the system
- The reference should be update in recent years.
Except the methodological references (i.e. reference:19:); Majority of the references have updated.
Reviewer 3 Report
Comments to the Authors
Manuscript ID: medicina-2517551
General remarks:
The manuscript “Investigation of the bioaccessibility of precursors of advanced glycation products in gluten-free products by using in vitro gastrointestinal system” compares the glyoxal (GO) and methylglyoxal (MGO) contents of gluten-free and gluten-containing bread, biscuits, cookies, and investigates their bioaccessibility in the in vitro gastrointestinal model. This is a very important aspect of GF products and diet not only celiac patients but also for healthy population on a GF diet so the present research could widen the existing knowledge in the field. However, there are some issues that should be addressed in the manuscript. Lots of technical errors such as extra space, missing dots, and dashes between the words (gluten-free, gluten-containing) are present in the manuscript and must be corrected. Furthermore, additional proximate composition analysis of the samples (moisture, ash, protein, lipid, starch content) should be included and confirmed/declined the influence of samples composition on the glyoxal (GO) and methylglyoxal (MGO) contents by conducting correlation analysis.
Abstract: is well written.
Introduction: Is well written.
Page 2, Lines 94 – 96 – Please add the exact precursors determined and methods used for that purpose as well as the exact GF product type in order to better focus the issue in the study aim.
Materials and methods:
Page 3, Line 109 – Please include the number of appropriate Table with this data in the supplementary material here.
Page 3, Line 119 – Please correct to 19 gluten containing products
Page 3, Line 120 – Please include furnisher, city and country of the used grinder and the obtained particle size.
Page 3, Line 122 – Please include furnisher, city and country of the used Ultra Turrax mixer and the used mixing speed and dispersion tool.
Page 4, Line 140 – Please indicate the temperature in the water bath.
Results and Discussion:
Page 6, Lines 210 – 213 – Please revise and correct this sentence because is unclear.
Page 6, Line 215 – It should be “…between gluten-free and gluten-containing food groups”. Please correct.
Page 7, Line 236 – Please correct “bakeried products” to baked products.
Page 7, Lines 227 – 239 and Page 8, Lines 240 – 252 – These sections of discussion are the same. Please revise and delete.
Page 8, Lines 259 – 260 – Unclear sentence, please improve.
Page 8, Lines 271 – 272 – Please revise and improve this sentence.
Page 8, Line 276 – Please correct “tendeny” to tendency.
Page 8, Line 285 – Please correct Tablo 3 to Table 3.
Page 9, Lines 298 – 308 – Since the unsaturated fatty acids and protein content are associated with increased GO and MGO levels their content in the investigated cookies should be determined and presented too.
Conclusion:
Page 12, Line 373 – Please correct inreteactions to interactions.
Table 2 – Please develop all the abbreviations used in tables and table titles, since a Table should be self-explainable.
Table 3, 4 – Same as for Table 2.
Figure 1 – Please translate text on figure to English.
Lots of technical errors such as extra space, missing dots, and dashes between the words (gluten-free, gluten-containing) are present in the manuscript and must be corrected.
Author Response
REVIEWER 3
General remarks:
The manuscript “Investigation of the bioaccessibility of precursors of advanced glycation products in gluten-free products by using in vitro gastrointestinal system” compares the glyoxal (GO) and methylglyoxal (MGO) contents of gluten-free and gluten-containing bread, biscuits, cookies, and investigates their bioaccessibility in the in vitro gastrointestinal model. This is a very important aspect of GF products and diet not only celiac patients but also for healthy population on a GF diet so the present research could widen the existing knowledge in the field. However, there are some issues that should be addressed in the manuscript. Lots of technical errors such as extra space, missing dots, and dashes between the words (gluten-free, gluten-containing) are present in the manuscript and must be corrected. Furthermore, additional proximate composition analysis of the samples (moisture, ash, protein, lipid, starch content) should be included and confirmed/declined the influence of samples composition on the glyoxal (GO) and methylglyoxal (MGO) contents by conducting correlation analysis.
- Thank you very much for your constructive feedbacks about the value of manuscript.
- In section that ‘’Instructions for Authors of Journal’’ there isn’t any technical information such as space, font , font size ..etc ( https://www.mdpi.com/journal/medicina/instructions). For this reson the technical errors have been corrected by use MEDICINA Microsoft Word Template Microsoft Word template.
- On the other hand, the point you draw attention to ''composition analysis of the samples''' has explained below:
Not only nutrient composition but also storage conditions and thermal processing of food affect the AGEs content1,2 The overall goal of in vitro gastrointestinal digestibility is to focus on determining how the AGE contents of foods differ from pre-digestive values and their bioaccessibility in the digestive environment 3-5 . The aim of this study is also similar. The sample size of the study is high. Therefore, the study budget is mainly devoted to the analysis of in vitro digestion parameters. Separate analysis could not be performed for each nutritional factor (except sugar analysis) affecting AGE content. These compositional factors affecting the formation of AGE; only the data obtained from the label information and with the literature information have discussed. However, these factors you mentioned are among the targets of our future work. In the works we will plan; all nutritional parameters affecting the formation of AGEs, storage and thermal processing conditions will be determined, analysed and recorded.Then analyzed in in vitro digestion environments. Thus, it is aimed to make correlations with product specific actual data. It is thought that this study will inspire future studiesIt is also emphasised in conclusion part.
1Tian, Z., Chen, S., Shi, Y., Wang, P., Wu, Y., & Li, G. (2023). Dietary advanced glycation end products (dAGEs): An insight between modern diet and health. Food Chemistry, 415, 135735.
2Uribarri, J., Woodruff, S., Goodman, S., Cai, W., Chen, X., Pyzik, R., Yong, A., Striker, Gary E. and Vlassara, H. (2010). Advanced glycation end products in foods and a practical guide to their reduction in the diet. Journal of the American Dietetic Association, 110(6), 911-916.
3 Serin, Y., Akbulut, G., UÄŸur, H., & Yaman, M. (2021). Recent developments in in-vitro assessment of advanced glycation end products. Current Opinion in Food Science, 40, 136-143.
4 Ozgolet, M., Yaman, M., Durak, M. Z., & Karasu, S. (2022). The effect of five different sourdough on the formation of glyoxal and methylglyoxal in bread and influence of in vitro digestion. Food Chemistry, 371, 131141.
5 Yaman, M., Demirci, M., Ede-Cintesun, E., Kurt, E., & Mızrak, Ö. F. (2022). Investigation of formation of well-known AGEs precursors in cookies using an in vitro simulated gastrointestinal digestive system. Food Chemistry, 373, 131451
Abstract: is well written.
Thank you very much.
Introduction: Is well written.
Thank you very much.
Page 2, Lines 94 – 96 – Please add the exact precursors determined and methods used for that purpose as well as the exact GF product type in order to better focus the issue in the study aim.
Thank you very much. The aim of the study have ordered again.
Materials and methods:
Page 3, Line 109 – Please include the number of appropriate Table with this data in the supplementary material here.
Thank you very much. It is added
Page 3, Line 119 – Please correct to 19 gluten containing products.
Thank you very much. It is corrected.
Page 3, Line 120 – Please include furnisher, city and country of the used grinder and the obtained particle size.
Thank you very much. It is added
Page 3, Line 122 – Please include furnisher, city and country of the used Ultra Turrax mixer and the used mixing speed and dispersion tool.
Thank you very much. It is added
Page 4, Line 140 – Please indicate the temperature in the water bath.
Thank you very much. It is added
Results and Discussion:
Page 6, Lines 210 – 213 – Please revise and correct this sentence because is unclear.
Thank you very much. It is revised and corrected.
Page 6, Line 215 – It should be “…between gluten-free and gluten-containing food groups”. Please correct.
Thank you very much. It is corrected.
Page 7, Line 236 – Please correct “bakeried products” to baked products.
Thank you very much. It is corrected.
Page 7, Lines 227 – 239 and Page 8, Lines 240 – 252 – These sections of discussion are the same. Please revise and delete.
Thank you very much it is deleted.
Page 8, Lines 259 – 260 – Unclear sentence, please improve.
Thank you very much; ıt is modified.
Page 8, Lines 271 – 272 – Please revise and improve this sentence.
Thank you very much; ıt is modified.
Page 8, Line 276 – Please correct “tendeny” to tendency.
Thank you very much. It is changed as ‘’ expectation’’
Page 8, Line 285 – Please correct Tablo 3 to Table 3.
Thank you very much. It is corrected.
Page 9, Lines 298 – 308 – Since the unsaturated fatty acids and protein content are associated with increased GO and MGO levels their content in the investigated cookies should be determined and presented too.
This part; has been explained under general point of the reviewer with references and justifications.
Conclusion:
Page 12, Line 373 – Please correct inreteactions to interactions.
Thank you very much. It is corrected.
Table 2 – Please develop all the abbreviations used in tables and table titles, since a Table should be self-explainable.
Thank you very much. It is corrected.
Table 3, 4 – Same as for Table 2.
Thank you very much. It is corrected.
Figure 1 – Please translate text on figure to English.
Thank you very much. Ä°t is translated.
Comments on the Quality of English Language
Lots of technical errors such as extra space, missing dots, and dashes between the words (gluten-free, gluten-containing) are present in the manuscript and must be corrected.
The manuscript has checked by proof and editing service and the certificate of editing upload to the system
Round 2
Reviewer 1 Report
The author has responded to the reviewer's comment point by point.
The author has responded to the reviewer's comment point by point.
Author Response
Thank you very much for your valuable feedback.
The revision of manuscript and the file of response has uploaded to the system.
Best regards;
Dear Editor;
Dear Reviewers;
Thank you very much for your spending time, careful review, and invaluable contribution to this manuscript.
On all reviewers' suggestions; we sent the manuscript to proof and editing service again for extensive English revision. The changes and corrections of editing services were shown in a uploded separate pdf file.
- TITLE
In order to prevent repetation the products word; ''gluten free-products '' were changed as gluten-free foods ( higlighted with yellow colour).
DISCUSSION PART
In order to clarify the flows in line 270-275 we added a comment that ''The proportional difference of amylose/amylopectin may be influence to digestive enzymes activity and indirectly the amount of AGEs that released from food''
Reviewer -3
Thank you very much for your contrubitions. We are happy with your suggestion that adding nutritional label information to Supplemantary Material. We added this information to ''Supplemantary Material Table S2'' .
We also added an comment related nutritional label information of samples in discussion part in line: 279-280 ''
-In all text; we changed the writing syle of gluten free word to ''gluten-free''.
-All references that relevant to the contents of the manuscript were checked .
Best regards;
Reviewer 2 Report
The author has responded to the reviewer's comment.
The author has responded to the reviewer's comment.
Author Response

(The authors gave the same response as above.)

Reviewer 3 Report
Please replase "gluten free" with "gluten-free". I suggest to revise whole manuscript to correct this issue. It is understandable that a complete proximate compositiom analysis is financialy mor consumable. However, the manuscript would be more ingsitful if the proximate composition found in the products nutritional labels is presented in supplementary material. I sugget to the Authors to do this since it would be beneficial for saike of comparation with results of further experiments planed by the Authors.
Minor corrections required.
Author Response

(The authors gave the same response as above.)
